

# Alzheimer's diseases in America, Europe, and Asian regions: a global genetic variation

Rahni Hossain[1], Kunwadee Noonong[1,2], Manit Nuinoon[1], Udom Lao-On[1,2], Christopher M. Norris[3], Pradoldej Sompol[3], Md. Atiar Rahman[1,4], Hideyuki J. Majima[1,2] and Jitbanjong Tangpong[1,2]

[1] School of Allied Health Sciences, College of Graduate Studies, Walailak University, Nakhon Si Thammarat, Thailand
[2] Research Excellence Center for Innovation and Health Product (RECIHP), School of Allied Health Sciences, Walailak University, Nakhon Si Thammarat, Thailand
[3] Department of Pharmacology & Nutritional Sciences, College of Medicine, University of Kentucky, Lexington, Kentucky, United States
[4] Department of Biochemistry and Molecular Biology, University of Chittagong, Chittagong, Bangladesh

Corresponding author
Jitbanjong Tangpong,
rjitbanj@wu.ac.th

## ABSTRACT

**Background:** Alzheimer's disease (AD) is one of the multifaceted neurodegenerative diseases influenced by many genetic and epigenetic factors. Genetic factors are merely not responsible for developing AD in the whole population. The studies of genetic variants can provide significant insights into the molecular basis of Alzheimer's disease. Our research aimed to show how genetic variants interact with environmental influences in different parts of the world.

**Methodology:** We searched PubMed and Google Scholar for articles exploring the relationship between genetic variations and global regions such as America, Europe, and Asia. We aimed to identify common genetic variations susceptible to AD and have no significant heterogeneity. To achieve this, we analyzed 35 single-nucleotide polymorphisms (SNPs) from 17 genes (ABCA7, APOE, BIN1, CD2AP, CD33, CLU, CR1, EPHA1, TOMM40, MS4A6A, ARID5B, SORL1, APOC1, MTHFD1L, BDNF, TFAM, and PICALM) from different regions based on previous genomic studies of AD. It has been reported that rs3865444, CD33, is the most common polymorphism in the American and European populations. From TOMM40 and APOE rs2075650, rs429358, and rs6656401, CR1 is the common investigational polymorphism in the Asian population.

**Conclusion:** The results of all the research conducted on AD have consistently shown a correlation between genetic variations and the incidence of AD in the populations of each region. This review is expected to be of immense value in future genetic research and precision medicine on AD, as it provides a comprehensive understanding of the genetic factors contributing to the development of this debilitating disease.

## INTRODUCTION

In 1907, Alois Alzheimer observed his patient, Auguste Deter, who had been experiencing memory loss, disorientation, hallucinations, and delusions. After her death at the age of 55, Alois Alzheimer reported on the clinical and pathological findings of her brain, which later became known as Alzheimer's disease (AD) (*Cipriani et al., 2011*). According to the World Health Organization (WHO), AD is a type of dementia that destroys brain cells and nerves, leading to memory loss, difficulties in thinking, and behavioral changes. AD accounts for 50–60% of all dementia cases (*Liu, Wang & Wang, 2021*). AD has two primary forms: early-onset Alzheimer's disease (EOAD), which affects about 5% of people and occurs in those under 60–65 years old, and late-onset Alzheimer's disease (LOAD), which affects about 95% of people and occurs in those over 60–65 years old. It is a common and life-threatening disease for the growing elderly population, especially those with LOAD (*Hollingworth et al., 2012*).

Epidemiological projections estimate that the global prevalence and consequences of AD in people aged 65 years or older will increase from 24 million to four times that number by 2050 (*Kumar et al., 2018*). AD occurrence within the general population seems to be between 5.5% and 9% over 6 months (*Schachter & Davis, 2000*). AD can start impacting the brain a decade before exhibiting any indications due to its three phases: mild or early-stage Alzheimer's, moderate or middle-stage Alzheimer's, and severe or late-stage Alzheimer's (*Sperling et al., 2011*). Two types of neuropathological lesions primarily characterize AD. These lesions include amyloid plaques that are made up of Aβ peptides. These peptides are formed due to proteolytic cleavage of amyloid precursor protein (APP) by β- and γ-secretase. The main products of this cleavage are $A\beta_{1-40}$ and $A\beta_{1-42}$. Neurofibrillary tangle (NFTs) accumulation of hyperphosphorylated Tau proteins is also essential in understanding the symptoms and severity of AD. This is due to the correlation between the location and density of tau NFT (*Ballardsych et al., 2011*; *Kang, Yh & Lee, 2017*). Three genes are responsible for EOAD, namely presenilin 1 (PSEN1), presenilin 2 (PSEN2), and APP, have been linked with the formation of AD through the involvement of $A\beta_{42}$ (*Vigneswaran et al., 2021*). The fourth gene, apolipoprotein E (APOE), has three common variants: APOE2, APOE3, and APOE4, coded by corresponding alleles (ε2, ε3, ε4). APOE ε4 is a significant risk factor for LOAD (*Huebbe & Rimbach, 2017*). Around 2% of the population that has AD carries two E4 alleles, which increases their risk by approximately 12 times. On the other hand, around 20–25% of the population has at least one copy of APOE4, which increases their risk of AD by about four times compared to those who have the more common APOE3/ε3 genotype (*Verghese, Castellano & Holtzman, 2011*). Patients with AD it is expected to experience Olfactory dysfunction (OD), which can be accompanied by cognitive impairment, neuropsychiatric symptoms, and difficulties with performing activities of daily living (*Yu et al., 2018*). When identifying risk and phenotypes for AD, it is important to consider genetic and non-genetic factors, including motor and behavioral factors. Understanding the changeability of clinical appearance is also crucial for tracking progression (*Montero-Odasso, Ismail & Camicioli, 2020*).

Genome-wide association studies (GWAS) have associated increased risk of LOAD with several genomic variants or single-nucleotide polymorphisms (SNPs) found in chromosomes 9, 10, 11, and 19, including rs2075650 TOMM40, rs6656401 CR1, rs11691237 BIN1, and rs3865444 CD33 (*Logue et al., 2011*; *Lambert et al., 2013*; *Chung et al., 2014*; *Omoumi et al., 2014*). Most studies and case controls performed individual meta-analyses to investigate the association between LOAD and several genetic variants or polymorphism. GWAS identified genes that disrupt lipid metabolism, immune response, and intracellular trafficking as potential mechanisms in LOAD (*Pottier et al., 2012*; *Harold et al., 2009*). Rare coding sequence variants in genes like APP, PSEN1, PSEN2, ADAM10, TREM2, and PLD3 have been found in patients with LOAD, indicating their contribution to the disease risk (*Vardarajan et al., 2015*). A study found 12 coding mutations in seven genes in three datasets, including seven autopsy-confirmed LOAD cases. Three rare coding mutations were observed in all three datasets: rs138047593 in BIN1, rs202178565 in EPHA1, and rs138650483 in MS4A6A (*Vardarajan et al., 2015*).

In 2013, a consortia group united under the International Genomics of Alzheimer's Project (IGAP). They conducted a mega-meta-analysis of their combined data, which included 17,008 cases of AD and 37,154 controls. The results were then validated in an independent sample of 8,572 Alzheimer's cases and 11,312 controls (*Lambert et al., 2013*). Studies have collectively discovered 19 new risk loci through single variant analyses that consistently meet the accepted $p$-value threshold for genome-wide significance ($p < 5 \times 10^{-8}$). As there is currently no cure for AD, the focus of medication is to decelerate its progression and address coexisting conditions like anxiety, depression, and sleep disorders (*Jin, 2015*). AD diagnosis can be attributed to certain brain parts, such as the cerebral cortex and hippocampus, which display decreased synaptic density and neuronal loss (*Reitz & Mayeux, 2014*). According to *Lanctôt et al. (2017)*, the Alzheimer's Association reports that individuals with Alzheimer's typically exhibit early signs of memory loss, difficulties with planning and problem-solving, challenges in completing familiar tasks, troubles with time and place recognition, issues with visual perception, communication deficits in both speech and writing, mood changes, and struggles with maintaining social connections (*Lanctôt et al., 2017*). Identifying the genetic risk factors of each population of the different regions provides information to understand the disease mechanism sensibly (*Lambert et al., 2010*). This scoping review examined the different studies on genetic variants and AD to identify the susceptible genetic variants or polymorphisms based on their region.

## MATERIALS AND METHODS

### Literature search and inclusion criteria

We searched PubMed (http://www.ncbi.nlm.nih.gov/PubMed) and Google Scholar (http://scholar.google.com) to query the articles cited to select all possible studies with keywords including "Alzheimer's disease," "genetic variant," "single-nucleotide polymorphism" and "SNP of Alzheimer's diseases" were used to search for relevant articles. The literature was searched for relevant articles from 2007 to 2022. As the genetic variants or SNPs are susceptible to AD individually in different populations, we aimed to combine and compare

**Table 1 Characteristics of included studies on the American population.**

| Gene | Variant/ polymorphism | Chromosome position | Major/minor alleles | Country | n cases | n controls | OR (95% CI) | p-Value | Study |
|---|---|---|---|---|---|---|---|---|---|
| CD33 | **rs3865444**[a] | 51,727,962 | C/A | USA | 572 | 1,340 | 0.93 [0.80–1.09] | – | *Lambert et al. (2013)* |
| BDNF | rs6265 | 26,467,157 | A/G | USA | 220 | 128 | 2.57 [1.19–5.55] | 0.0327 | *Huang et al. (2007)* |
| | rs11030104 | 26,471,758 | C/T | | | | | 0.0534 | |
| | rs2049045 | 26,481,482 | C/G | | | | | 0.0226 | |
| CR1 | rs6656401 | – | G/A | Canada | 580 | 524 | 1.29 [0.96–1.74] | 0.085 | *Omoumi et al. (2014)* |
| | rs3818361 | | | | | | | | |
| TOMM40 | rs2075650 | – | A/G | | | | 2.58 [2.04–3.25] | 0.026 | |
| BIN1 | rs7561528 | – | G/A | | | | 1.55 [1.06–2.27] | 0.025 | |
| CD33 | **rs3865444**[a] | – | G/T | | | | 0.62 [0.48–0.80] | 0.001 | |
| BIN1 | rs11691237 | – | T (minor) | USA | 513 | 496 | 1.52 [1.11–2.09] | 0.0098 | *Logue et al. (2011)* |
| | rs11685593 | – | T (minor) | | | | 1.66 [1.13–2.45] | 0.0098 | |
| | rs7585314 | – | T (minor) | | | | 0.75 [0.62–0.91] | 0.0030 | |
| CLU | rs2279590 | – | T (minor) | | | | 1.41 [1.03–1.95] | 0.034 | |
| | rs9331926 | – | G (minor) | | | | 1.96 [1.11–3.48] | 0.020 | |
| EPHA1 | rs4595035 | – | T (minor) | | | | 1.25 [1.06–1.47] | 0.0094 | |
| MS4A | rs10792258 | – | T (minor) | | | | 0.79 [0.66–0.95] | 0.010 | |
| ABCA7 | rs3764647 | – | G (minor) | | | | 1.32 [1.07–1.63] | 0.0087 | |
| CD33 | rs10419982 | – | A (minor) | | | | 1.38 [1.15–1.65] | 0.00054 | |
| PICALM | rs17148827 | – | C (minor) | | | | 2.01 [1.19–3.40] | 0.0089 | |
| | rs12795381 | – | C (minor) | | | | 0.49 [0.29–0.84] | 0.0086 | |
| CD2AP | rs9349407 | – | C (minor) | Jacksonville | 492 | 922 | 1.10 [0.91–1.33] | 0.34 | *Carrasquillo et al. (2011)* |
| EPHA1 | rs11767557 | – | | | 501 | 957 | 0.86 [0.70–1.06] | 0.17 | |
| ARID5B | rs2588969 | – | A (minor) | | 495 | 928 | 1.04 [0.88–1.23] | 0.63 | |
| CD33 | **rs3865444**[a] | – | | | 492 | 920 | 0.82 [0.68–0.98] | 0.03 | |
| CD2AP | rs9349407 | – | C (minor) | Rochester | 313 | 1,600 | 0.88 [0.70–1.09] | 0.24 | |
| EPHA1 | rs11767557 | – | | | 309 | 1,572 | 0.89 [0.69–1.13] | 0.33 | |
| ARID5B | rs2588969 | – | A (minor) | | 307 | 1,604 | 1.12 [0.92–1.37] | 0.26 | |
| CD33 | **rs3865444**[a] | – | | | 312 | 1,577 | 0.88 [0.72–1.08] | 0.23 | |
| CD2AP | rs9349407 | – | C (minor) | Autopsy | 285 | 100 | 0.98 [0.65–1.47] | 0.92 | |
| EPHA1 | rs11767557 | – | | | 307 | 99 | 0.66 [0.43–1.02] | 0.06 | |
| ARID5B | rs2588969 | – | A (minor) | | 307 | 102 | 1.24 [0.86–1.79] | 0.24 | |
| CD33 | **rs3865444**[a] | – | | | 298 | 97 | 0.84 [0.57–1.24] | 0.39 | |
| MTHFD1L | rs11754661 | 151,248,771 | A (minor) | USA | 931 | 1,104 | 2.10 [1.67–2.64] | $1.90 \times 10^{-10}$ | *Naj et al. (2010)* |
| ABCA7 | rs3764650 | – | – | USA | 151 | 157 | 1.01 [0.58–1.75] | – | *Hollingworth et al. (2011)* |
| MS4A | rs610932 | – | – | | | | 0.88 [0.64–1.22] | – | |
| | rs670139 | – | – | | | | 1.08 [0.78–1.49] | – | |

**Notes:**
[a] with bold color indicates the polymorphism rs3865444, CD33 is frequent in the American population.
ABCA7, ATP-binding cassette, subfamily A (ABC1), member 7; ARID5B, AT-rich interaction domain 5B; BIN1, bridging integrator 1; BDNF, brain derived neurotrophic factor; CD2AP, CD2-associated protein; CD33, myeloid-associated antigen CD33; CLU, clusterin; CR1, complement component receptor 1; EPHA1, ephrin type-A receptor 1; MS4A, the membrane-spanning 4A gene cluster; MTHFD1L, methylenetetrahydrofolate dehydrogenase (NADP+ dependent) 1 like; PICALM, phosphatidylinositol binding clathrin assembly protein; TOMM40, translocase of outer mitochondrial membrane 40; SNP, single-nucleotide polymorphism. OR, odds ratio; CI, confidence interval.

**Table 2 Characteristics of included studies on the European population.**

| Gene | Variant/ polymorphism | Chromosome position | Major/minor alleles | Country | n cases | n controls | OR (95% CI) | p-Value | Study |
|---|---|---|---|---|---|---|---|---|---|
| TFAM | rs2306604 | – | A/G | Sweden | 406 | 318 | 0.997 [0.6–1.44] | 0.03 | *Belin et al. (2007)* |
| CD33 | **rs3865444**[b] | 51,727,962 | C/A | UK | 490 | 1,066 | 1.00 [0.84–1.19] | – | *Lambert et al. (2013)* |
| | | | | Sweden | 797 | 1,506 | 0.99 [0.87–1.12] | – | |
| | | | | Spain | 2,121 | 1,921 | 0.94 [0.85–1.04] | – | |
| | | | | Belgium | 878 | 878 | 0.95 [0.79–1.14] | – | |
| | | | | Finland | 422 | 562 | 1.08 [0.89–1.30] | – | |
| | | | | Germany | 972 | 2,378 | 1.00 [0.89–1.13] | – | |
| | | | | Greece | 256 | 229 | 0.79 [0.52–1.20] | – | |
| | | | | Italy | 1,729 | 720 | 1.12 [0.97–1.30] | – | |
| | | | | Hungary | 125 | 100 | 0.94 [0.60–1.47] | – | |
| | | | | Austria | 210 | 829 | 1.08 [0.82–1.42] | – | |
| CD2AP | rs9349407 | – | C (minor) | Norway | 324 | 519 | 0.81 [0.62–1.06] | 0.13 | *Carrasquillo et al. (2011)* |
| EPHA1 | rs11767557 | – | | | 338 | 548 | 0.94 [0.71–1.24] | 0.67 | |
| ARID5B | rs2588969 | – | A (minor) | | 338 | 543 | 1.05 (0.83–1.33) | 0.69 | |
| CD33 | **rs3865444**[b] | – | | | 327 | 541 | 0.89 [0.70–1.14] | 0.37 | |
| CD2AP | rs9349407 | – | C (minor) | Poland | 468 | 181 | 1.04 [0.77–1.42] | 0.79 | |
| EPHA1 | rs11767557 | – | | | 557 | 169 | 0.93 [0.66–1.31] | 0.67 | |
| ARID5B | rs2588969 | – | A (minor) | | 473 | 185 | 0.91 [0.68–1.20] | 0.49 | |
| CD33 | **rs3865444**[b] | – | | | 467 | 187 | 1.00 [0.72–1.37] | 0.99 | |
| ABCA7 | rs3764650 | – | – | France | 2,025 | 5,328 | 1.21 [1.08–1.37] | – | *Hollingworth et al. (2011)* |
| MS4A | rs610932 | – | – | | | | 0.93 [0.86–1.00] | – | |
| | rs670139 | – | – | | | | 1.06 [0.99–1.14] | – | |

**Notes:**
[b] with bold color indicates the polymorphism rs3865444, CD33 is frequent in the European population.
TFAM, transcription factor A, mitochondrial.

all the populations of different regions and genetic variants to show the most susceptible one from each region. Database searches yielded potentially relevant articles, plus 80 studies identified from reference lists. After removing duplicates and screening titles and abstracts, full texts of 40 articles were retrieved, and 17 studies met all the inclusion criteria. The studies that met the following criteria were included: (1) the link between a genetic variant or polymorphism and AD; (2) the case-control studies conducted on people; and (3) the considered polymorphism for each region. Ultimately, we selected 17 independent studies, including 128,044 samples (51,550 cases and 76,494 controls) from American, European, and Asian populations, as described in Tables 1–3.

## Exclusion criteria

We carried out a methodical evaluation approach, which included eliminating 20 duplicate publications from an original collection of 100 databases. We were left with a list of 80 papers after this first selection that had the potential to be eligible based on our study goals. A total of 40 of these 80 publications were meticulously examined and found to be out of alignment with the predetermined study objectives, hence being disqualified from further

**Table 3 Characteristics of included studies on the Asian population.**

| Gene | Variant/polymorphism | Chromosome position | Major/minor alleles | Country | n cases | n controls | OR (95% CI) | p-Value | Study |
|---|---|---|---|---|---|---|---|---|---|
| APOC1 | rs4420638 | 45,422,946 | A/G | Korea | 400 | 605 | 3.53 [2.76–4.52] | $1.8 \times 10^{-23}$ | Chung et al. (2014) |
| TOMM40 | **rs2075650**[c] | 45,395,619 | A/G | | | | 3.19 [2.45–4.17] | $1.2 \times 10^{-17}$ | |
| APOE | **rs429358**[d] | 45,411,941 | T/C | Korea | 290 | 554 | 4.24 [3.01–5.96] | $1.23 \times 10^{-16}$ | Chung et al. (2013) |
| | **rs2075650**[c] | 45,395,619 | A/G | | | | 3.57 [2.51–5.06] | $1.23 \times 10^{-12}$ | |
| PICALM | rs677909 | 85,757,589 | A/G | | | | 0.63 [0.49–0.81] | 0.00036 | |
| TOMM40 | **rs2075650**[c] | – | A/G | China | 787 | 791 | 1.52 [1.19–1.94] | 0.016 | Ma et al. (2013) |
| | rs157580 | – | G/A | | | | 0.72 [0.57–0.92] | 0.001 | |
| APOE | **rs429358**[d] | 50,103,781 | T/C | Japan | 547 | 715 | 3.84 [3.06–4.81] | 0.0018 | Takei et al. (2009) |
| SORL1 | rs11218343 | – | C (minor) | | 891 | 844 | 0.83 [0.75–0.92] | $3.8 \times 10^{-4}$ | Miyashita et al. (2013) |
| | rs3781834 | – | G (minor) | | | | 0.74 [0.66–0.84] | $7.3 \times 10^{-7}$ | |
| CD33 | rs3865444 | – | G/T | China | 190 | 193 | 0.48 [0.35–0.66] | $3.23 \times 10^{-6}$ | Deng et al. (2012) |
| MS4A6A | rs610932 | – | A/C | | | | 0.622 [0.47–0.83] | 0.001 | |
| PICALM | rs3851179 | – | G/A | | 2,022 | 4,209 | 0.88 [0.78–0.99] | 0.03 | Liu et al. (2013) |
| B1N1 | rs744373 | – | – | China, Japan, India, and Turkey | 24,771 | 35,324 | 1.10 [1.02–1.19] | 0.01 | Almeida et al. (2018) |
| CR1 | **rs6656401**[e] | – | – | | | | 1.26 [1.02–1.56] | 0.04 | |
| CR1 | **rs6656401**[e] | – | – | India | 544 | 324 | 1.34 [1.05–1.71] | 0.009 | Biffi et al. (2012) |

**Notes:**
[c] with bold color indicates the polymorphisms rs2075650 APOE and TOMM40.
[d] rs429358 APOE.
[e] rs6656401 CR1 are common in the Asian population.
SORL1, sortilin related receptor 1.

consideration. The remaining 40 papers, initially thought to be pertinent to our research goals, had 23 eliminated since they had no significant data for our study. Following this meticulous filtering procedure, we devised a final list of 17 publications that satisfied the requirements to be included in our scoping review study.

## Data extraction

The following information was extracted from each study: (1) the name of the first author; (2) the ethnicity of the studied population; (3) the odds ratio (OR; 95% CI), chromosome position, and major/minor alleles of each polymorphism; (4) cases and control of each population. All the detailed information is described in Tables 1–3.

## RESULTS

### Search results

A flow chart that shows the articles included in this study is shown in Fig. 1. Our search returned 100 articles, 83 of which were eliminated due to failure to meet our inclusion criteria, leaving 17 articles used in the analysis. Some articles contained more than one study, as shown in our analysis.

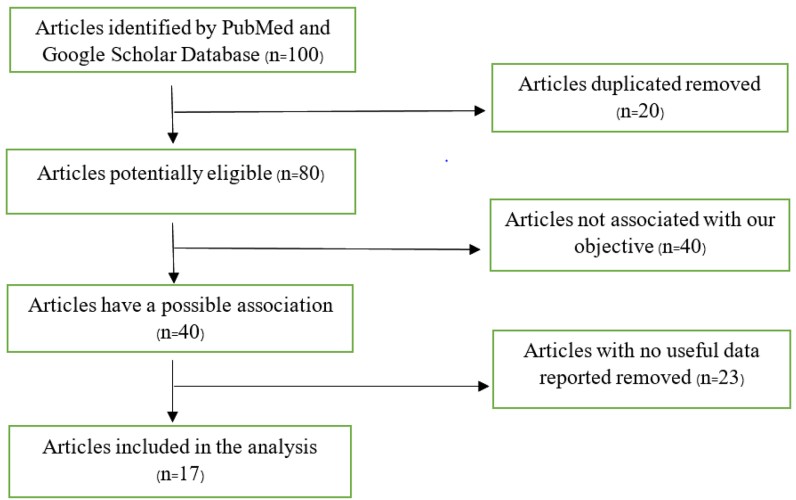

**Figure 1** **Flow chart of articles found on PubMed and Google Scholar database.**

## Characteristics of the included studies

Of the 17 studies reviewed, 30% of the articles focused on the American population study examining the association between AD and specific factors. A total of 20% of the articles were based on European population studies, while 50% investigated Asian population studies. Each region explored different polymorphisms to identify the susceptibility to AD among their respective populations.

## Results of qualitative syntheses

The study conducted in America and Europe revealed that different populations possess similar genes, namely ABCA7, ARID5, CD33, CD2AP, EPHA1, and MS4A, with polymorphism rs3764650, rs2588969, rs3865444, rs9349407, rs11767557, and rs610932, rs670139 respectively, responsible for causing AD in specific regions. Tables 1 and 2 display these findings. Previous studies showed that the American population has identified several frequently investigated polymorphisms susceptible to AD. These include rs3865444, rs17148827, rs6265, rs12795381, rs11030104, rs9349407, rs2049045, rs11767557, rs6656401, rs2588969, rs3818361, rs11754661, rs2075650, rs3764650, rs7561528, rs610932, rs11691237, rs670139, rs11685593, rs10419982, rs7585314, rs3764647, rs2279590, rs10792258, rs9331926, and rs4595035. We found that for association studies in the American population, CD33 rs3865444 is the most common. A total of 2,254 cases and 4,458 control with CD33 rs3865444 among American populations in the USA (*Lambert et al., 2013*), Canada (*Omoumi et al., 2014*), Jacksonville, Rochester, and Autopsy (*Carrasquillo et al., 2011*) have been studied. The results demonstrated a susceptible association with AD, as shown in Table 1. We prepared a forest plot in Fig. 2 showing the average odds ratio of 0.81 for AD associated with CD33 rs3865444 from individual regions.

In addition, the European population has shown that CD33 rs3865444 is the most commonly experimented polymorphism out of all the other variations, namely rs2306604,
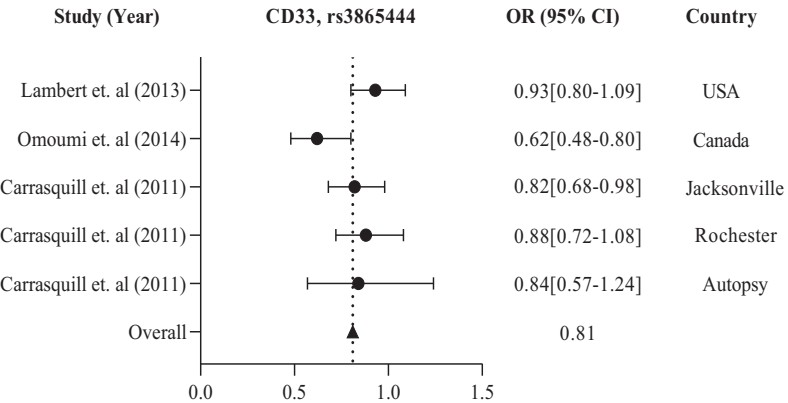

**Figure 2 The forest plot of the odds ratio of common polymorphism rs3865444, CD33 for the American population.** The data are presented by GraphPad Prism Data Editor for Windows, version 9.0 (GraphPad Software Inc, San Diego, CA, USA). Values are expressed by the average collective odds ratio of common rs3865444, CD33 polymorphism of America region (*Lambert et al., 2013*; *Omoumi et al., 2014*; *Carrasquillo et al., 2011*).

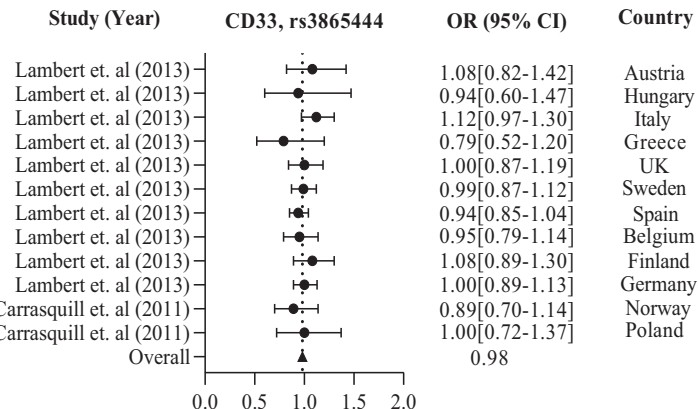

**Figure 3 The Forest Plot of the odds ratio of common polymorphism rs3865444, CD33 for the European population.** The data are presented by GraphPad Prism Data Editor for Windows, version 5.0 (GraphPad Software Inc, San Diego, CA, USA). Values are expressed by the average collective odds ratio of common rs3865444, CD33 polymorphism of Europe region (*Lambert et al., 2013*; *Carrasquillo et al., 2011*).

rs9349407, rs11767557, rs2588969, rs3764650, rs610932, and rs670139. Among the European countries, such as the UK, Sweden, Spain, Finland, Germany, Greece, Italy, Belgium, Hungary, Austria, Poland, and Norway, there is a high frequency of CD33 rs3865444 in a large population of 8,794 cases and 10,917 controls, which is associated with AD (*Carrasquillo et al., 2011*; *Lambert et al., 2013*). The details are shown in Table 2. We observed the combined odds ratio of CD33 rs3865444 using a forest plot and found that the odds ratio averaged 0.98, as depicted in Fig. 3.

Further in Asian region with 12 total polymorphisms were rs4420638, rs2075650, rs429358, rs67909, rs157580, rs11218343, rs3781834, rs3865444, rs610932, rs3851179, rs744373 and rs6656401 that are associated with a risk factor for AD. From these polymorphisms, researchers in Korea and China have studied rs2075650 polymorphism

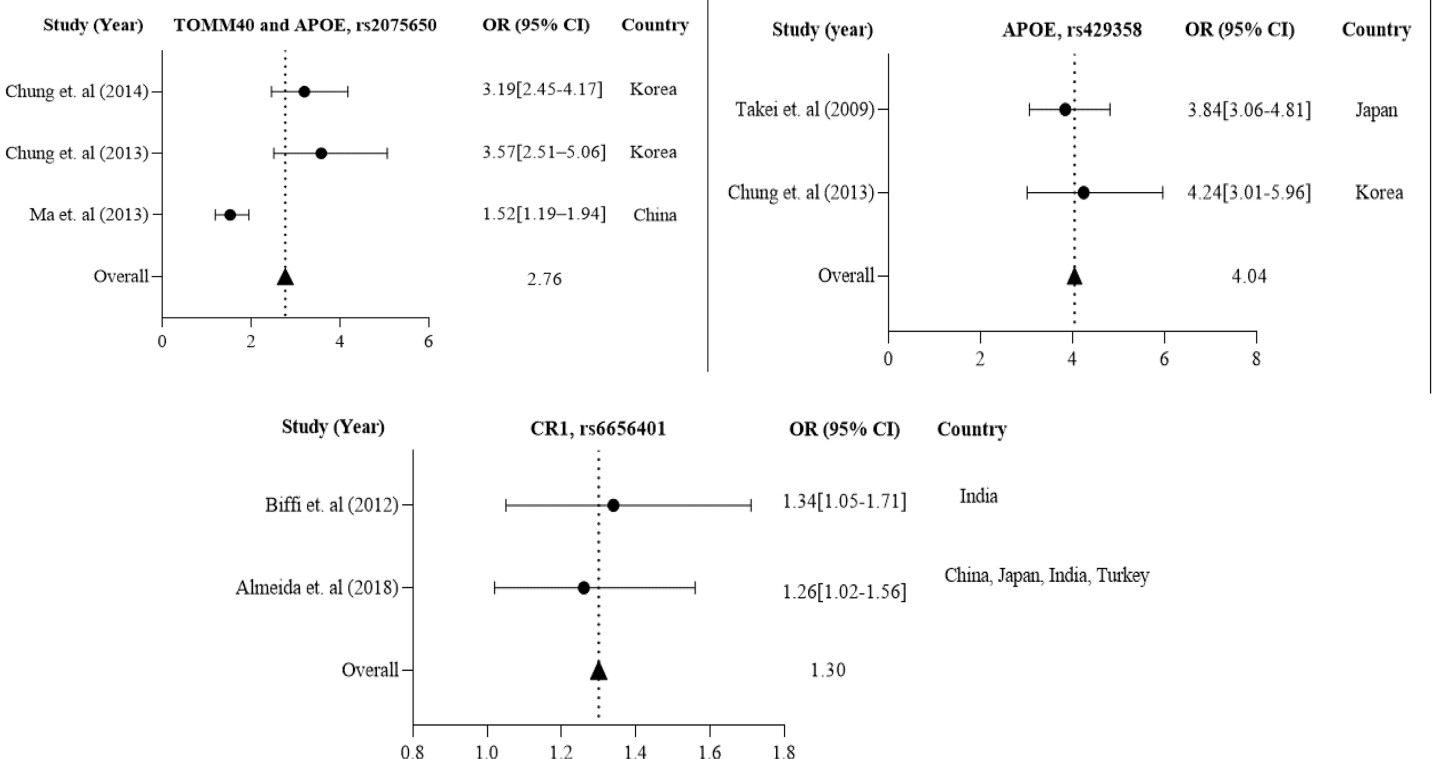

**Figure 4** **The Forest Plot of the odds ratio of common polymorphisms rs2075650, TOMM40, and APOE; rs429358, APOE, and rs6656401, CR1 for the Asian population.** The data are presented by GraphPad Prism Data Editor for Windows, version 5.0 (GraphPad Software Inc, San Diego, CA, USA). Values are expressed by the average collective odds ratio of the Asia region's common rs2075650, TOMM40, and APOE; rs429358, APOE, and rs6656401, CR1 (*Chung et al., 2014*; *Ma et al., 2013*; *Takei et al., 2009*; *Chung et al., 2013*; *Biffi et al., 2012*; *Almeida et al., 2018*; *Lambert et al., 2013*; *Carrasquillo et al., 2011*).

from both TOMM40 and APOE gene(s), which has shown to have a susceptibility to AD with their population (1,477 cases and 1,950 controls) with a *p*-value of 0.016 (*Chung et al., 2013*; *Ma et al., 2013*; *Chung et al., 2014*).

APOE rs429358 has been studied in 837 AD cases and 1,269 controls from Korea (*Chung et al., 2013*) and Japan (*Takei et al., 2009*). CR1 rs6656401 is common in 25,315 AD cases and 35,648 controls from a large population in India (*Biffi et al., 2012*), China, Japan, and Turkey (*Almeida et al., 2018*). The forest plot of Fig. 4 shows that the average odds ratio of APOE and TOMM40 rs2075650, APOE rs429358, and CR1 rs6656401 is 2.76, 4.04, and 1.30, respectively.

## Association of dementia with AD and prevalence of dementia in different regions

Dementia is a syndrome characterized by a progressive decline in memory, other mental skills, and behavior. It is brought on by several disorders that impact cerebral structures and functioning. Impaired capability to carry out daily tasks compromises autonomy and the ability to live independently, leading to dependence and care demands. Between 60% and 80% of dementia cases are thought to be caused by AD, making it the most prevalent subtype of dementia (*Mayeux & Stern, 2012*; *Nowrangi, Rao & Lyketsos, 2011*).
a.

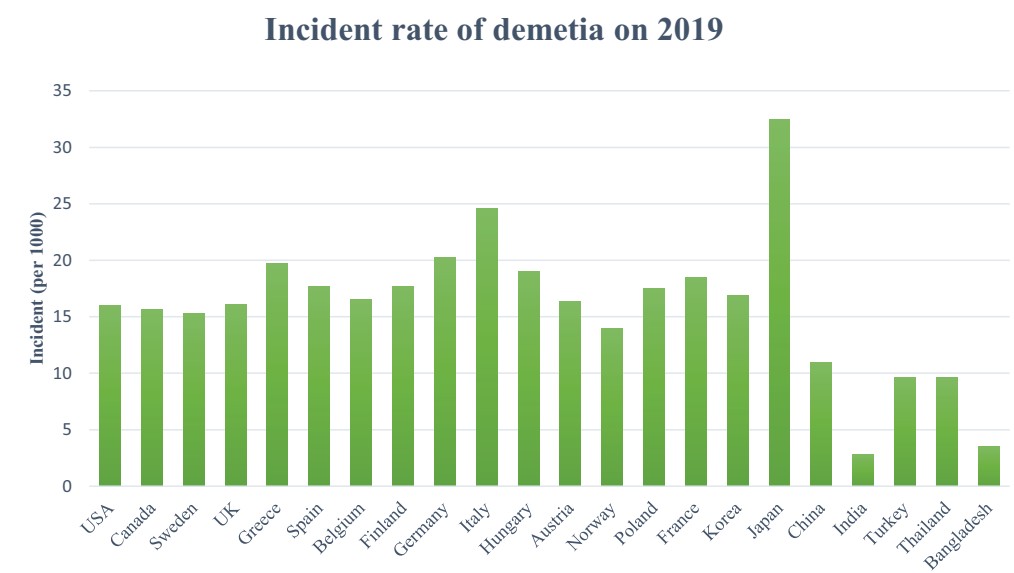

### Incident rate of demetia on 2019

b.

### Number of Dementia cases in 2019 and 2050 according the regions

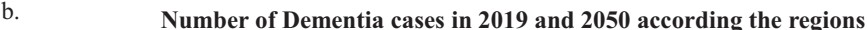

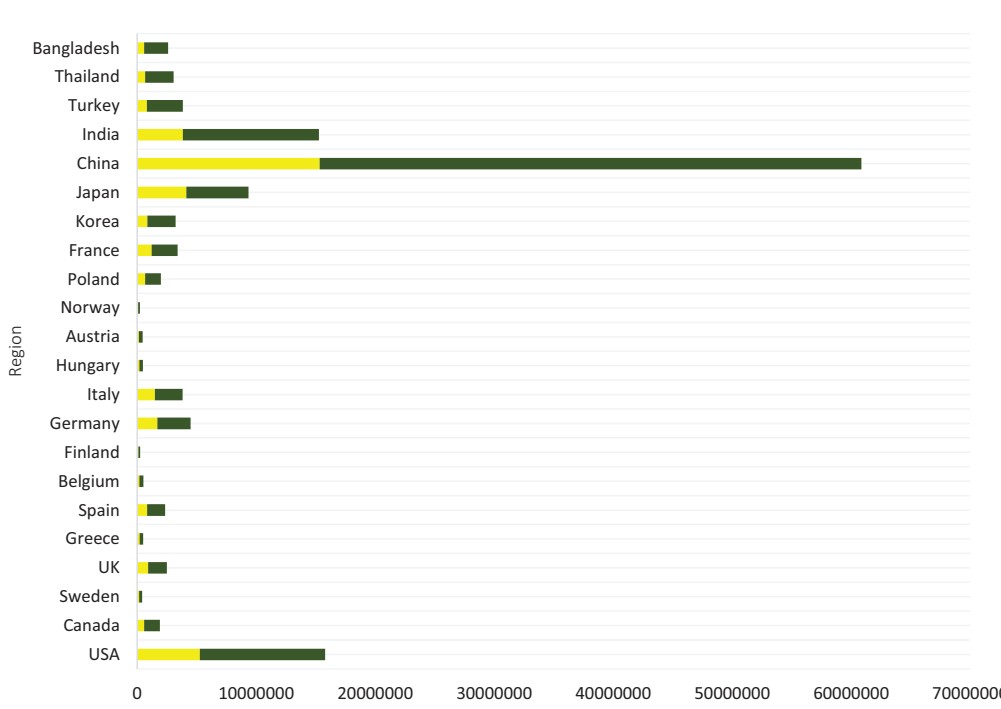

**Figure 5** **(A)** The incident rate of dementia in 2019 (per 1,000), **(B)** the number of all dementia cases between 2019 and 2050 globally and by world regions.

The number of people with dementia is anticipated to double every 20 years. It is estimated that before 2005, the occurrence of dementia among individuals aged 60 and above worldwide was 3.9%. The prevalence rate of dementia varies across different regions,

with Africa having the lowest rate of 1.6%, China and the Western Pacific regions having a rate of 4.0%, Latin America having a rate of 4.6%, Western Europe having a rate of 5.4%, and North America having the highest rate of 6.4% (*Ferri et al., 2005*). According to a study, there could be a significant increase in the number of people who have dementia worldwide in the next three decades. The study estimated that from the 57.4 million cases recorded in 2019, the number could rise to 152.8 million by 2050. The uncertainty interval for this estimate was 95%, ranging from 50.4 million to 65.1 million cases in 2019 and 130.8 million to 175.9 million cases in 2050 (*Nichols et al., 2022*). From 2019 to 2050, there is a significant increase in the projected number of cases of all ages living with dementia in all regions, and the incident rate of dementia in 2019 (per 1,000) is shown in Fig. 5A. The Global Burden of Diseases, Injuries, and Risk Factors Study (GBD) 2019 (*Murray et al., 2020*) has calculated the projected number of dementia cases in 2050 based on three risk factors (high body-mass index, high fasting plasma glucose, and smoking). The study has used relative risks and forecasted risk factor prevalence to predict the GBD risk-attributable prevalence in 2050 globally, as well as by world region and country (*Nichols et al., 2022*). In the USA, the percentage of people with Alzheimer's dementia for those aged 65 to 74 was 27.0%, for those aged 75 to 84 was 37.2%, and for those aged 85 and older was 35.7% in 2020 (*Rajan et al., 2021*). The country with the highest number of Alzheimer's patients is China, with 15,330,045 cases in 2019 and a projected 45,538,093 cases in 2050, as indicated in Fig. 5B. Table 3 shows that in China, TOMM40, CD33, MS4A6A, PICALM, BIN1, and CR1 polymorphisms are the genes responsible for most cases of AD.

## DISCUSSION

An investigation was conducted to identify the susceptibility gene(s) for LOAD in American, European, and Asian populations through a case-control association study using SNPs. Figure 6 displays the investigational polymorphisms of AD in different populations according to their respective regions. The study found that 17 genes for 35 polymorphisms are individually used for population studies of susceptibility to AD in various regions. The study focused on the following genes: APOE, CD33, BDNF, CR1, TOMM40, B1N1, CD2AP, EPHA1, ARID5B, MTHFD1L, TFAM, APOC1, PICALM, MS4A6A, CLU, SORL1, ABCA7, and PICALM. The study also showed that rs3865444, rs17148827, rs6265, rs12795381, rs11030104, rs9349407, rs2049045, rs11767557, rs6656401, rs2588969, rs3818361, rs11754661, rs2075650, rs3764650, rs7561528, rs610932, rs11691237, rs670139, rs11685593, rs10419982, rs7585314, rs3764647, rs2279590, rs10792258, rs9331926, rs4595035, rs2306604, rs610932, rs670139, rs429358, rs67909, rs157580, rs11218343, rs3781834, rs3851179, rs744373, and rs6656401 polymorphisms are used for investigation of American, European, and Asian populations. The study included 128,044 samples, with 51,550 cases and 76,494 controls, from American, European, and Asian populations to show the frequent investigational polymorphism according to the region. CD33 rs3865444 is commonly found in populations of American and European descent and has been linked to AD. The mitochondrial genome encodes 13 subunits for electron-transfer chains, two ribosomal

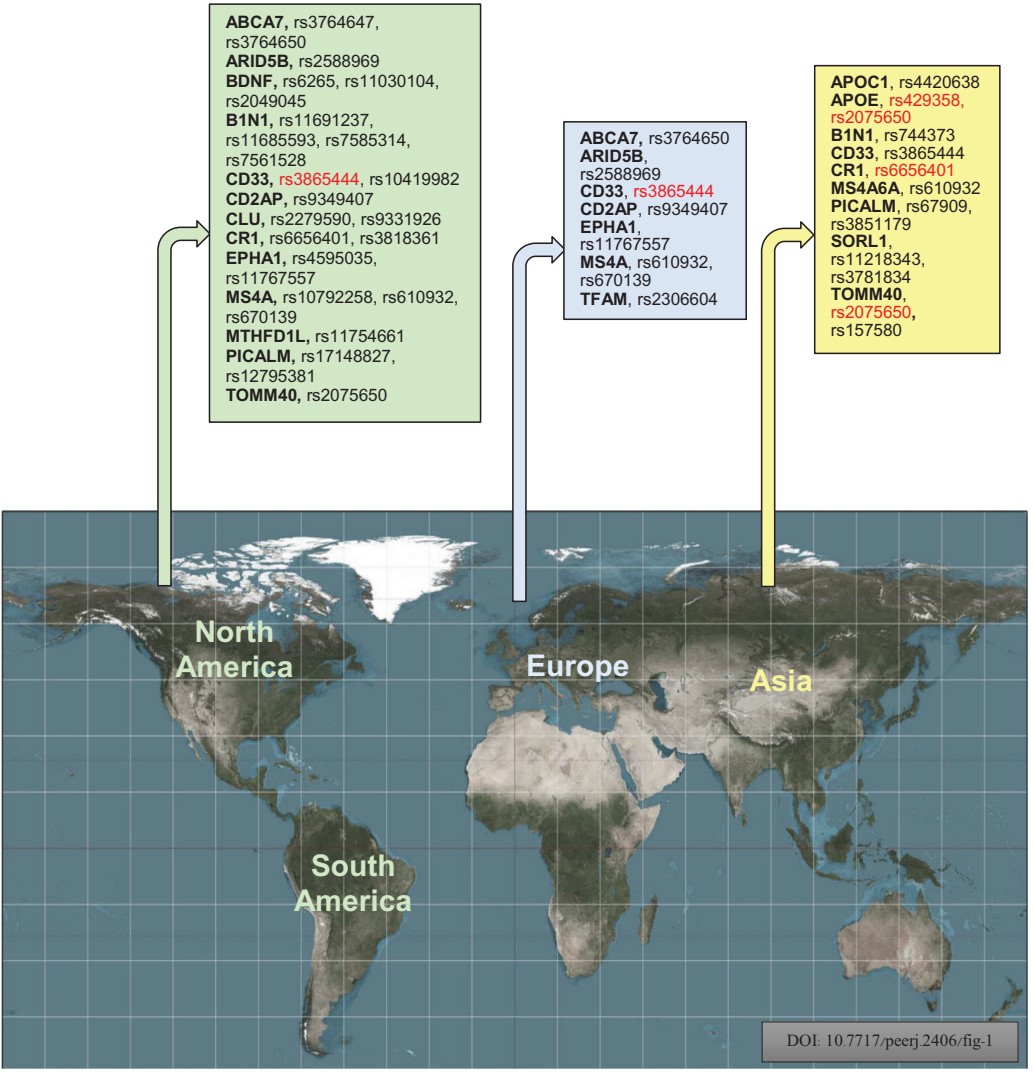

**Figure 6 Geographically, investigational polymorphisms of Alzheimer's disease of the different populations according to their regions (Red color indicates the common polymorphisms in America, Europe, and Asian populations) (*Bouckaert, 2016*).**

RNAs, and 22 transfer RNAs. Despite having fewer genes than the nuclear genome, it plays a vital role in energy production and cellular function (*Kang, Kim & Hamasaki, 2007*). The transcription of mtDNA, replication, damage sensing, and DNA repair are controlled by TOMM40 and TFAM mitochondrial genes. Mutations in mtDNA can decrease the ability to produce ATP, negatively affecting the function of muscles, neurons, and other cells by impairing energy supply (*DiMauro & Schon, 2003*; *Ekstrand et al., 2004*; *Kang & Hamasaki, 2005*). Increasing evidence shows reduced mitochondrial function plays a major role in AD (*Orth & Schapira, 2001*).

The increased risk of AD associated with the CD33 locus in American and European populations is likely due to the risk haplotype's greater inclusion of exon 2 in CD33 mRNA (*Raj et al., 2014*). The Canadian population has shown a significant *p*-value of 0.001 for the

G/T alleles of major and minor variants. For the Chinese population in the Asian region, the major and minor alleles A/G of TOMM40 rs2075650 have a significant $p$-value of 0.016. A polymorphism called CR1 rs6656401 is currently being studied in a large population of 25,315 cases across India, Turkey, Japan, and China. Our study suggests that each region's genetic variation or polymorphism depends on the patient's background, lifestyle, habits, and history. The GWAS report has linked BIN1 and CR1 genes to the risk factor of AD pathogenesis through C allele rs744373 and A allele rs6656401 polymorphisms (*Rajan et al., 2021*). Numerous studies on genetic association have confirmed that the APOE E4 allele is a significant risk factor for LOAD. Our research indicates that the APOE E4 allele plays a role in AD development and is associated with specific polymorphisms. Specifically, the APOE E4 allele directly regulates brain lipid metabolism and synaptic functions through APOE receptors (*Bu, 2009*). APOE is positioned on chromosome 19q13.32 and encodes a protein involved in lipid homeostasis and cholesterol metabolism. It plays a crucial role in regulating fibrillogenesis and Aβ clearance, thereby maintaining Aβ homeostasis. The risk of developing AD is higher in individuals with the ancestral E4 isoform than in other isoforms. Approximately 200,000 years ago, a mutation occurred at residue 112 of APOE, resulting in the E3 isoform with a cysteine residue instead of an arginine residue. This mutation protected cytotoxic domain interactions (*Xu et al., 2004*; *Suri et al., 2013*). Tables 1 and 2 show that the rs3865444 A and T allele was less common in individuals with AD. Conversely, the central C and G alleles were associated with an increased risk of AD, with OR values of 0.81 and 0.98, respectively, as indicated in Figs. 2 and 3. Another polymorphism, rs2075650, revealed that the significant A allele was more susceptible to AD than the minor G allele, with an OR value of 2.76, as shown in Table 3 and Fig. 4.

It is worth considering that Asian and African countries may yield different results due to age distribution, life expectancy after developing AD, genetic differences, diets, and physical exertion norms. The risk of AD is linked to various disorders such as hypertension, diabetes, obesity, and dyslipidemia, as well as social and environmental factors such as head trauma, physical activity, diet, socioeconomic status, and education level (*Gatz et al., 2006*; *Imtiaz et al., 2014*). There is a strong correlation between the prevalence rates of AD and the supply of fat and total calories, with respective correlation coefficients of 0.932 and 0.880 (*Pan et al., 2020*). In the future, genetic analysis will enable the assessment of polygenic risk for AD. In contrast, neuropsychological assessment will aid in identifying the accurate targeting pathology through multiple biomarkers for more precise treatment of the disease (*Bondi, Edmonds & Salmon, 2017*). In certain developing countries, the differences in the level of exposure to environmental risk factors can be attributed to low levels of cardiovascular risk (*Hendrie et al., 2001*) and hypolipidemia (*Chandra et al., 2001*) in individuals or both. However, in developing countries, other risk factors, such as anemia, which is more prevalent in rural India, are associated with AD (*Pandav et al., 2004*). As AD is mainly an illness later in life, it is more common in developed countries, where the prevalence is higher. The age-specific prevalence of AD almost doubles every 5 years after age 65 (*Lopez & Kuller, 2019*). Many individuals above the age of 65 in developed countries face varying levels of dementia, with dementia-related

symptoms and signs typically appearing only in those aged 85 and above (*Corrada et al., 2008*). Caregivers or care communities of individuals with Alzheimer's and related dementias provide care for an extended period compared to those caring for individuals with other conditions, according to the Centers for Disease Control and Prevention (*Centers for Disease Control and Prevention, 2019*). The COVID-19 pandemic has led to reduced healthcare and treatment for AD patients, with 50% of nursing homes serving as care communities for AD or other dementia patients (*Harris-Kojetin et al., 2019*).

As of December 2021, COVID-19 had claimed the lives of over 141,000 people living in long-term care communities (*Center for Medicare & Medicaid Services, 2020*). Of those who had Medicare fee-for-service (FFS) coverage and were hospitalized due to COVID-19 up to August 2021, 32% had a diagnosis of dementia (*Center for Medicare & Medicaid Services, 2021*). A recent study conducted a GWAS meta-analysis for AD, identifying two novel disease-associated loci on chromosome 3 and fine-mapping nine loci across multiple populations. The findings highlight the importance of multi-ancestry representation in exploring genetic factors influencing AD risk (*Lake et al., 2023*). AD risk is associated with age, family history, and genetics. A lack of diverse genetic data has hindered progress, but emerging insights from different ethnicities are allowing for a more comprehensive understanding of AD genetics and potential personalized approaches (*Reitz et al., 2023*). The present study aims to identify the most significant polymorphisms associated with AD in different regions of the world. The variations in polymorphisms are highly influenced by ethnic and geographical factors. Our findings underscore the importance of examining the genetic basis of AD across various populations and locations. It is crucial to gain insights into the genetic mechanisms underlying AD to facilitate the development of targeted interventions and personalized treatment approaches. While this scoping study provides valuable information on the genetic variations linked to AD in various ethnic groups, it has some limitations. It mainly focuses on the frequently studied polymorphisms in American, European, and Asian populations without considering any variations in less-researched regions that may have specific genetic risk factors for AD. Additionally, the study does not factor in other variables such as environmental, lifestyle, and epigenetic factors that can interact with genetic variations and significantly influence an individual's susceptibility to AD. Therefore, a comprehensive understanding of AD risk should consider a broader range of factors.

## Future direction and perspective

AD genetics require further research to address the limitations identified in this scoping review. To enhance the accuracy of genetic risk assessment for AD, future studies should aim to include a more diverse range of people, especially those from underrepresented locations. By doing so, a more comprehensive understanding of the global genetic landscape of AD susceptibility can be obtained. To better understand the complex relationship between genetics and other risk factors, further research is needed to investigate gene-gene and gene-environment interactions. Integrating multi-omics data from genomes, transcriptomics, proteomics, and epigenomics can lead to a more complete knowledge of the pathophysiology and risk of AD. To develop personalized treatment

strategies for AD that can be applied clinically, future studies should focus on identifying a person's genetic profile. Such tailored interventions may be able to halt or slow down the progression of the disease.

## CONCLUSION

This review summarizes the genetic variations associated with AD in American, European, and Asian populations. The study identifies several widespread variants linked to AD risk, such as CD33 rs3865444, TOMM40 rs2075650, and APOE rs429358. It is crucial to understand that AD is a complex and multifaceted disease influenced by both genetic and environmental factors. The identified genetic variations account for only a small fraction of the total genetic risk for AD. To gain a clearer understanding of the genetic basis of AD, future studies should continue to investigate its genetic causes in different population groups, considering the interplay between genes and other risk factors. A better understanding of the genetic foundation of AD can help develop more tailored and specialized strategies for early identification, prevention, and treatment of this debilitating neurodegenerative disorder through precision medicine.

## ABBREVIATION

| | |
|---|---|
| **ABCA7** | ATP-binding cassette, subfamily A (ABC1), member 7 |
| **APOC1** | apolipoprotein C1 |
| **APOE** | apolipoprotein E |
| **ARID5B** | AT-rich interaction domain 5B |
| **BIN1** | bridging integrator 1 |
| **BDNF** | brain-derived neurotrophic factor |
| **CD2AP** | CD2-associated protein |
| **CD33** | myeloid-associated antigen CD33 |
| **CI** | confidence interval |
| **CLU** | clusterin |
| **CR1** | complement component receptor 1 |
| **EPHA1** | ephrin type-A receptor 1 |
| **MS4A** | the membrane-spanning 4A gene cluster |
| **MTHFD1L** | methylenetetrahydrofolate dehydrogenase (NADP+ dependent) 1 like |
| **PICALM** | phosphatidylinositol binding clathrin assembly protein |
| **SORL1** | sortilin related receptor 1 |
| **TOMM40** | translocase of outer mitochondrial membrane 40 |
| **TFAM** | transcription factor A, mitochondrial |
| **SNP** | single-nucleotide polymorphism |
| **OR** | odds ratio |
### Funding
The authors received no funding for this work.

### Competing Interests
The authors declare that they have no competing interests.

### Author Contributions

- Rahni Hossain conceived and designed the experiments, performed the experiments, analyzed the data, prepared figures and/or tables, authored or reviewed drafts of the article, approved the final draft of the manuscript, and approved the final draft.
- Kunwadee Noonong conceived and designed the experiments, performed the experiments, analyzed the data, prepared figures and/or tables, authored or reviewed drafts of the article, approved the final draft of the manuscript, and approved the final draft.
- Manit Nuinoon conceived and designed the experiments, performed the experiments, analyzed the data, prepared figures and/or tables, authored or reviewed drafts of the article, approved the final draft of the manuscript, and approved the final draft.
- Udom Lao-On conceived and designed the experiments, performed the experiments, analyzed the data, prepared figures and/or tables, authored or reviewed drafts of the article, approved the final draft of the manuscript, and approved the final draft.
- Christopher M. Norris conceived and designed the experiments, performed the experiments, analyzed the data, prepared figures and/or tables, authored or reviewed drafts of the article, approved the final draft of the manuscript, and approved the final draft.
- Pradoldej Sompol conceived and designed the experiments, performed the experiments, analyzed the data, prepared figures and/or tables, authored or reviewed drafts of the article, approved the final draft of the manuscript, and approved the final draft.
- Md. Atiar Rahman conceived and designed the experiments, performed the experiments, analyzed the data, prepared figures and/or tables, authored or reviewed drafts of the article, approved the final draft of the manuscript, and approved the final draft.
- Hideyuki J. Majima conceived and designed the experiments, performed the experiments, analyzed the data, prepared figures and/or tables, authored or reviewed drafts of the article, approved the final draft of the manuscript, and approved the final draft.
- Jitbanjong Tangpong conceived and designed the experiments, performed the experiments, analyzed the data, prepared figures and/or tables, authored or reviewed drafts of the article, resource, approved the final draft of the manuscript, and approved the final draft.

## Data Availability

The raw data are available in the Supplemental File.

## Supplemental Information

Supplemental information for this article can be found online at http://dx.doi.org/10.7717/peerj.17339#supplemental-information.

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
