# Peer review of "Alzheimer’s diseases in America, Europe, and Asian regions: a global genetic variation"

_PeerJ, doi:10.7717/peerj.17339_

## Round 0.1 · original submission · Major Revisions

Please refer to the reviewers' comments and revise accordingly.

**Language Note:** The review process has identified that the English language must be improved. PeerJ can provide language editing services - please contact us at copyediting@peerj.com for pricing (be sure to provide your manuscript number and title). Alternatively, you should make your own arrangements to improve the language quality and provide details in your response letter. – PeerJ Staff

·

Basic reporting

Hossain et al. is a very important work giving a review of SNPs in the American, European, and Asian populations. However, there are a few suggestions that could justify this work and make it clearer.

Firstly, the English language should be improved to ensure that an international audience can clearly understand your text. Some examples where the language could be improved include lines 30-34, 38-39, 72-74, 77-81 – the current phrasing makes comprehension difficult. I suggest you have a colleague who is proficient in English and familiar with the subject matter review your manuscript or contact a professional editing service.

Secondly, in this review, some papers from as early as 1994 are referenced, which is about 30 years old. Only those old papers should be used that are landmark papers. Please use more recent references for those papers that have such alternatives.

Thirdly, although I understand that this is an important review, please justify clearly in the manuscript how this review is different and more important than those already in the literature and what new is being offered by this work.

Experimental design

In line 120, you mention the collection of 100 databases. Is this something that is a source somewhere already or you handpicked these 100 databases? If it is a single source, please add that to the reference list, and if they were handpicked, please provide the list of the databases used as a supplemental file.

Validity of the findings

No comment

Additional comments

Please see below a line-by-line commentary:

Line 38-39 - All the research found a link between genetic variations and AD in the populations 39 of each region (n cases, n controls) – the n cases n controls information is not necessary here, unless there is more to be added here
Line 54 - In the general population, AD's 6-month prevalence appears to be 5.5% to 9% 55 (Gao, Hendrie et al. 1998) – what does six-month prevalence mean?
Line 59 – definition of APP here instead of line 65
Line 63 – central component means?
Line 64 - has been associated with 3 autosomal dominants, deterministic – unclear
Line 69 - The population (2%) carries two E4 alleles, - please consider using “2% of the population with AD carries two E4 alleles”
Line 74-76 - Any manuscript should not contain “we” and this feels more of a future consideration for this type of research instead of being a basis for this work, and the closing parenthesis on line 76 is replaced by 0, probably a typographical error.
Line 80 – the years in references are look smaller
Line 83 – please reference IGAP
Line 88 - it slows the progression of AD with other – what is it?
Line 92 – please reference Alzheimer's Association
Line 107 – Instead of putting in the dates of actual search, put in the years you searched for in your query, for example, the literature was searched for relevant articles in the years 2018 through 2023.
Line 114 – what is considered as a “considerable polymorphism”? Did you mean to say “considered”?
Line 115 – seventeen should be replaced with 17, since the number is what has been used before
Line 141-142 – please use past tense for all the methods used
Line 145 – The entire section “Results of Qualitative Syntheses” has many English language related errors, especially after the long lists of rsIDs that you mention. The language is not clear in such cases and please check the entire manuscript for such errors.
Line 155 - CD33 rs3865444 is for trial in the – clinical trial? Please mention. Also, the English language is not clear.
Line 173-174 – Reference years look smaller
Line 182-185 – please mention the year for this information, which year was this global prevalence estimated in?
Line 187 – Please refer to the World Alzheimer 187 Report 2019 correctly.
Line 190 – please refer to the CHAP, US census report.
Line 190-194 – something missing?
Line 191 - please refer to the GBD
Line 195 - 2020 the percentage of people with Alzheimer’s dementia in the United States is 27.0% of people – please use past tense
Line 198-201 – something missing?
Line 204 - something missing?
Line 220 - 37 introns fewer genes that encode 13 subunits of the electron-transfer chain, 2 ribosomal RNAs, - introns fewer genes means?
Lines 230-232 – not clear
Line 240 – mention the reference 40.’
Line 263 - anemia found to be associated with AD in rural India will be more prevalent in developing – consider the use of “is” instead of “will be”
Line 273 - because of less treatment and health care during pandemics. – the use of “pandemics” makes it confusing: is this a comment for only COVID or multiple pandemics, please change accordingly
Line 279 - The review – if this point is for this work, please use “this” instead of “the”. “The” refers to any review in my opinion
Line 305 – But it's crucial to understand that AD is a multifaceted, complicated illness caused - No line should start with “But”
Line 563: Typographical error
In legends for tables 1 and 2, please mention what the bold rsIDs and the superscripts a and b mean, similar to the footnote mentioned after table 3.
Table 2, first page: the CD2AP major/minor allele is not completely visible
The legends mentioned right before the figures 2,3 and 4 are incomplete, please make sure that these legends and the legends section (Lines 530-563) match

Reviewer 2 ·

Basic reporting

Overall, this is a paper properly written with clear and professional language. However, I still suggest the author to double-check the grammar and the terminology.
This paper cited enough references with proper background and enough figures to support the statements. I hope that the author can provide a supplementary text file that categorizes the paper reviewed into different regions or SNPs to better help with the reader's research.
The review will attract broad interest within the scope of the journal, especially within the genetics and neurology field.
This field is constantly reviewed. Recently there was a paper published in Nature Reviews 'A global view of the genetic basis of Alzheimer disease' in this field. However, this review is short, and clear with useful information about SNP. Many papers reviewed in this manuscript are related to the Asian population which provides a unique point of view of unveiling this field.
The introduction is clear, but I would like to suggest more information about how mutations potentially impact the molecular biology pathway or mechanism of Alzheimer’s disease to let more audience interested in this article.

Experimental design

The survey methodology appears to be thorough and aims for comprehensive coverage of the subject. The authors also introduced the inclusion and exclusion of the literature search. There are enough sources cited and the review is well organized

Validity of the findings

There is enough evidence provided to support the findings. The conclusion also summarizes the findings well but could be improved in suggesting specific future research directions. The discussion can be improved by discussing potential biases or limitations of the studies included in the review if possible. The paper should also improve the connection and transition between paragraphs in the discussion and between the discussion and the conclusion.
Overall this is a well-structured and well-written paper.

Reviewer 3 ·

Basic reporting

The presented scoping review aims to aggregate and summarize current literature on GWAS findings among American, Eurpopean and Asian populations and highlight the genetic variants among them. While this is a review that would benefit the community, I have several concerns regarding the review.

1. As the review focuses on aggregating and summarizing genetic findings across several populations, it is essential to include current state of literature in this area of study. However, such background is lacking in the Introduction section. Current state of literature needs to be detailed and it must be made clear how the presented review differs in methodology as well as in its primary focus.

2. Recently published, highly relevant articles were not included in the introduction or discussion sections to discuss how the findings of this review relate to the existing literature. For example: (1) Reitz et al. 2023, https://doi.org/10.1038/s41582-023-00789-z; (2) Lake et al. 2023, https://doi.org/10.1038/s41380-023-02089-w.

3. Prevalence of dementia in different regions as well as number of their projected cases in 2050 are presented in results and discussion sections, but methods section does not explain how they were identified/derived. Further, their relevance to main focus of the study is unclear and they do not appear to be relevant to the review.

4. First sentence of discussion section -- "A case-control was carried out association study using SNPs to determine the susceptibility gene(s) for LOAD in American, European, and Asian populations" -- appears to wrongly suggest authors performed the case-control association study.

5. Tables 1, 2 and 3 are often missing crucial information such as chromosomal position.

6. Methodology is incomplete when stating the criteria used to exclude 23 articles.

Experimental design

No comment. See basic reporting section.

Validity of the findings

No comment. See basic reporting section.

---

## Round 0.2 · Minor Revisions

Please refer to the reviewers' comments for revision.

·

Basic reporting

Clear and unambiguous professional English was used throughout, but there are still some language related errors. The review has a lot of important information but a justification of why this review adds to the literature and is different than reviews already available is missing. A sentence saying that this work focuses on populations that have not been studied in detail before or in recent past, or this paper caters to a different population, is strongly suggested.

Experimental design

No comment.

Validity of the findings

No comment.

Reviewer 2 ·

Basic reporting

This review matches the interest and is within the scope of the journal. This review provides a good reason and introduces the importance of reviewing the process of Alzheimer’s disease. The English language in this paper is significantly improved. However, checking with Grammarly can not guarantee the correctness of English usage for papers. Thus, I suggest the authors have a careful proof read to futhre fix potential problems.

Experimental design

The author provides a good description of the survey methodology with the cited paper provided. I suggest the author add reasons for each paper removed from the 100 papers.

Validity of the findings

The finding is valid. However, the authors start to discuss "dementia" from line 200 without introducing the connection between dementia and AZ. Because those two concepts are different, I strongly suggest authors do not mix the usage of those two words and rewrite this part of the paper.

---

## Round 0.3 · accepted · Accept

No further comments from the reviewers.